# Alcoholic Acute Pancreatitis, a Retrospective Study about Clinical Risk Factors and Outcomes—A Seven-Year Experience of a Large Tertiary Center

**DOI:** 10.3390/biomedicines12061299

**Published:** 2024-06-12

**Authors:** Deniz Gűnșahin, Andrei Vicențiu Edu, Mihai Radu Pahomeanu, Tudor Ștefan Mitu, Andreea Irina Ghiță, Anamaria Simona Odorog, Carmen Monica Preda, Lucian Negreanu

**Affiliations:** 1Faculty of Medicine, Carol Davila University of Medicine and Pharmacy, B-dul. Eroii Sanitari, 8, 050474 Bucharest, Romaniapreda_monicaa@yahoo.com (C.M.P.);; 2Gastroenterology Department, Emergency Clinical Hospital Bucharest, Calea Floreasca, 8, 014461 Bucharest, Romania; 3Internal Medicine & Gastroenterology Department, University Emergency Hospital of Bucharest, Splaiul Independenței, 169, 050098 Bucharest, Romania; 4Gastroenterology Department, Fundeni Clinical Insititute, Soseaua Fundeni, 258, 022328 Bucharest, Romania

**Keywords:** acute pancreatitis, alcohol, ICU, recurrence, risk factors, outcome, pseudocyst, cost

## Abstract

(1) Background: Alcohol consumption is one of the main causes of acute pancreatitis. (2) Material and Methods: In this unicentric retrospective cohort study, we selected 1855 patients from the Bucharest Acute Pancreatitis Index (BUC-API) who presented with acute pancreatitis. We investigated correlations between Alcoholic Acute Pancreatitis (AAP) and the rate of complications, cost, length of hospitalization and rate of recurrence. (3) Results: We found a moderately strong association between AAP and recurrence (*p* < 0.01) and observed that the disease is likelier to evolve with pseudocysts and walled-off necrosis than other forms of AP. Patients with AAP are less likely to have a morphologically normal pancreas than patients suffering from AP of other causes (*p* < 0.01), but a low probability of requiring intensive care unit admission (*p* < 0.01) significantly lowers daily cost (Md = 154.7 EUR compared to Md = 204.4 EUR) (*p* < 0.01). (4) Conclusions: This study’s data show that patients with AAP have a greater rate of pseudocyst occurrence, lower intensive care unit admittance rate and lower cost of hospitalization than patients with AP of other causes. Typical Sketch: A middle-aged male tobacco smoker with recurrent AP, lower risk of in-hospital mortality and complications such as pseudocysts; treated in a gastroenterological ward and discharged at-will.

## 1. Introduction

With a prevalence rate ranging from 8.8% in Israel [1] to 47% in Finland [2], alcohol consumption, defined by the Center for Disease Control (CDC) [3] as consuming more than 40 g of pure alcohol per day, is one of the main etiologies of AP worldwide.

In-hospital mortality in AAP depends on the type of study selected, but varies between 1.1% [4] and 8.4% [5]. The highest mortality rate related to AAP has been reported in Russia (2.7/100,000 people) [6].

In the current study, we sought to address a gap in AAP research related to prevention strategies by improving medical knowledge related to modifiable risk factors and effective prevention strategies. We aimed to evaluate the associations between AAP and several clinical and demographic attributes related to known causes of AP. Our main aim was to investigate the possible significant links with ICU admittance. A secondary aim was to establish links between the following variables and AAP: age, sex, outcome at discharge, ward of origin, morphology, rurality, tobacco smoking, daily cost of hospitalization, severity and length of stay (LoS). These factors are useful not only in determining those at risk of developing AAP in the general population but also in finding primary prophylaxis measures and optimizing healthcare systems that treat this illness.

## 2. Materials and Methods

### 2.1. Bucharest Acute Pancreatitis Index (BUC-API)

This large retrospective cohort study included patients selected from the BUC-API registry, a unicentric registry of 2039 cases of AP, recurrent AP and acute-on-chronic pancreatitis.

Using the BUC-API registry was approved by the IRB at the University Emergency Hospital of Bucharest (Spitalul Universitar de Urgență București). Informed consent on behalf of the patients was obtained before hospitalization. This study follows the STROBE guidelines and the ethical guidelines of the 1975 Declaration of Helsinki.

Although the cases analyzed in this study represent 35 of the 42 counties in Romania, most of the registry’s cases were treated in southern Romania in Bucharest and in Giurgiu, Ilfov and Teleorman counties.

Cases from 1 June 2015 to 1 April 2022 were selected from the electronic health records (EHRs) at the University Emergency Hospital of Bucharest using the ICD-10 codes K85, B26.3 and B25.2 as consecutive hospitalizations.

Initially, we found 2520 cases, which were then screened by trained medical staff. Of these, we excluded 52 miscoded cases and 426 duplicates caused by software processing errors. An additional 184 cases were found to be acute-on-chronic pancreatitis cases that had not been excluded previously because of a lack of ICD-10 codes regarding this condition, which is differentiated from AP and RAP by the following: known previous diagnosis of chronic pancreatitis, concerning imaging findings (calcifications in pancreatic parenchyma, pancreatic duct dilation and/or proofs of exocrine pancreatic insufficiency). The co-existence of a tumoral mass within the pancreas (primary carcinoma or secondary determination) represented an exclusion criterion from the registry. In total, 1855 consecutive cases of AP in 1618 unique patients were included in the current study. To the best of our knowledge, this is the largest registry study of pancreatitis performed to date in our country.

Founded in 1978, the University Emergency Hospital of Bucharest is the largest acute-care tertiary teaching hospital in Romania. It accommodates 1099 beds and has both gastroenterological and abdominal surgery departments.

### 2.2. Case Selection and Definitions

Details regarding the patients’ characteristics relevant to this study are found in Table 1. In order to control for statistical bias, for any patient that had multiple episodes in our registry, we took into consideration only the last chronological episode.

Patients that had no data regarding daily cost, morphology, residency or tobacco usage were excluded from the statistical analysis of that particular subject.

Severity and morphology were stratified as in the Revised Atlanta Classification [7].

Tobacco smoking was considered former if it had ceased at least 4 weeks before hospital admission; however, quantifying how many packs of cigarettes were smoked by all active and former smokers was impossible to determine from the data available in EHRs.

Etiology was stratified as in “Sleisenger and Fordtran’s Gastrointestinal and Liver Diseases”, 10th edition [8].

To determine the alcoholic cause of AP, we applied the IAP/APA guidelines [9] regarding interviews of patients with AAP (to assess heavy drinking). Interview was conducted by the caring physician. Where doubt was retained relating to the alcoholic etiology after interview the caring physician sought additional indirect proofs like (a majority of them should have been positive simultaneously): deRitis ratio >2.0, macrocytosis absent, B12 deficiency or other hematological disease that could explain it, gamma-glutamyl transferase high in absence of any hepatic disease that could explain it, paper-money skin sign, essential tremor unexplained by other neurological or psychiatric pathology, Dupuytren contracture and alcohol withdrawal signs after 24 h from admission.

To make better comparisons with the AAP group, all cases of AP not of alcoholic were combined to form the other-cause AP (OAP) group.

Patients suffering from acute-on-chronic pancreatitis (calcification in pancreas, dilation of pancreatic ducts and/or proofs of exocrine pancreatic insufficiency) were excluded from the study population before any analysis.

A patient’s first attack of AP during the selected registry timeframe was considered any case that did not mention previous hospitalization in our hospital or prior episodes of AP.

The outcome of each case was documented as reported by the case physician. The cost of hospitalization was calculated in RON and reported in EUR by conversion with the exchange rate published by Romanian National Bank on 10 May 2024, respectively 1.00 EUR = 4.97 RON.

### 2.3. Statistical Analysis

The database used in this study was organized using Excel 2019 and Google Docs. To analyze the cohort’s general characteristics (Table 1), frequency tests were used. Pearson’s chi-square test and Cramer’s V (phi test) were used to evaluate the correlations between two categorical variables. The Mann–Whitney U test was used to assess correlations between continuous and other categorical variables. All statistical tests were performed using IBM SPSS ver. 29.0.0.0. Statistical significance was achieved at *p* < 0.05. A third decimal place was reported only when the *p*-value was between 0.04 and 0.05.

## 3. Results

### 3.1. Population Characteristics

We found 1618 consecutive patients with AP in our registry. Of the patients in our registry, over half were male (n = 918, 56.7%). The patients’ median age was 59.0 years (IQR = 26.0). Relatively equal numbers of admittances to the gastroenterology and surgery wards, which lasted a median LoS of 7.0 days (IQR = 6.0), were noted.

We established a median daily cost of 177.2 EUR. At discharge, the majority of patients (n = 1329, 82.1%) achieved a good outcome (healed or ameliorated). Most (n = 821, 50.7%) experienced a mild course of disease, with few cases necessitating ICU admittance (n = 169, 10.4%). Concerning morphology, as per the Revised Atlanta Classification, we discovered that 38.7% (n = 626) developed interstitial pancreatitis.

Of the patients, 503 (31.1%) were classified as having AAP. Of the remaining patients, we found a rate of idiopathic AP at 15.6% (n = 253).

Most of our subjects lived in urban areas (n = 1159, 71.6%). Some cases (n = 355, 21.9%) had data regarding tobacco smoking status; most of those reported active smoking (n = 250, 15.5% of total population).

### 3.2. Recurrence

We performed a chi-square test of independence to identify a correlation between AAP and recurrence and found a significant moderate association between the two variables: χ^2^ (1, n = 1618) = 107.8, *p* < 0.01. Thus, AAP cases are likelier to be associated with recurrence (see Table 2 and Figure 1).

### 3.3. Outcome at Discharge

A chi-square test of independence was performed to assess the relationship between etiology and outcome at discharge, revealing a significant relationship between the two variables: χ^2^ (4, n = 1618) = 34.5, *p* < 0.01. In addition, a phi test (φ = 0.15) revealed a moderate magnitude association. We observed that patients with AAP were likelier to be discharged at-will but less likely to be transferred and to suffer from in-hospital mortality (see Table 2).

### 3.4. Gender

To evaluate the affiliation of gender with AAP, we performed a chi-square test of independence, finding that men are more prone to developing AAP than women: χ^2^ (1, n = 1618) = 307.0, *p* < 0.01, φ = 0.44 (see Figure 2).

### 3.5. Age

A Mann–Whitney U test was conducted to determine whether associations between age and AAP existed. The results revealed that AAP develops at a significantly younger age (M = 52.0 years, SD = 13.0) than other forms of AP (M = 60.3 years, SD = 18.1) (U = 365,910.5, *p* < 0.01) (see Figure 3).

### 3.6. Severity

To determine an association between etiology and severity, we ran a chi-square test of independence, finding no significant differences between the two groups: χ^2^ (2, n = 1618) = 2.5, *p* = 0.28.

### 3.7. ICU Admittance

To determine a correlation between etiology and ICU necessitation, we conducted a chi-square test of independence, which showed a lower probability of patients’ with AAP requiring ICU admittance, although the association was weak: χ^2^ (1, n = 1618) = 17.1, *p* < 0.01, φ = 0.1 (see Figure 4).

### 3.8. Morphology

A chi-square test of independence was carried out to determine if there were any associations between etiology and morphology. The results showed medium magnitude statistically significant associations: χ^2^ (5, n = 1114) = 29.6, *p* < 0.01, φ = 0.16. Thus, AAP is likelier to evolve with pseudocysts and WON, and patients are less likely to have a morphologically normal pancreas (see Figure 5).

### 3.9. Cost of Hospitalization

To assess if there were any cost discrepancies regarding treating AAP, we performed a Mann–Whitney U test, which revealed a significantly lower daily expense for patients with AAP (M = 311.1 EUR, SD = 2878.9) than patients with OAP (M = 552.4 EUR, SD = 4029.1) (U = 280,087.0, *p* < 0.01).

### 3.10. Type of Caring Ward

To determine whether patients with AAP were admitted to a particular ward, we used a chi-square test of independence. The results showed substantial differences between the two variables and a strong magnitude association: χ^2^ (1, n = 1618) = 333.8, *p* < 0.01, φ = −0.45. In our study, patients with AAP were likelier to be admitted to the gastroenterological ward.

### 3.11. Urban or Rural Residency

To assess the correlation between residency and type of AP, we used a chi-square test of independence, finding no significant link: χ^2^ (1, n = 1604) = 1.44, *p* = 0.23.

### 3.12. Tobacco Usage

A chi-square test was performed to determine any association between patients with AAP and their use of tobacco products. A medium magnitude association strength was found: χ^2^ (2, n = 355) = 24.1, *p* < 0.01, φ = 0.26. Patients with AAP were likelier to be active tobacco smokers.

### 3.13. Number of Days of Stay (LoS)

To determine a correlation between LoS and etiology, we ran a Mann–Whitney U test, which revealed no significant differences (U = 280,428.5, *p* = 0.99).

## 4. Discussion

AAP remains the most common etiology of AP in many areas of the world, especially Eastern Europe [6] and our country [10].

In this study, we looked at whether there are any significant differences between AAP and all other etiologies of AP in terms of recurrence of acute episodes, evolution, hospitalization costs and complications.

Because of the recent increased interest regarding the risk factors of recurrent AP [11,12,13], researchers have found an association between RAP and AAP. This is of great importance because RAP frequently progresses into chronic pancreatitis, an independent risk factor for pancreatic cancer. Most probably this effect is not likely as much related to the toxicity of ethanol but to its pattern of usage as most other causes of AP, mainly biliary, find their resolution if not in the index admission soon after, alcoholism is a chronic state in which the patient exposes itself repeatedly to the risk factor and as such expose themself to other episodes of AP. This discussion should be dealt by prospective studies with long follow-up periods.

Compared with OAP the in-hospital mortality rate was statistically significant lower. Still, we documented a higher in-hospital mortality rate (4.4%) in patients with AAP than comparable studies’ rates of 1.3% [14], 2.63% [15] and 1.0% [16]; however, these studies’ outcome data originated in the USA, so a bias may be present related to differences in wealth, healthcare systems and medical access.

While we could not find studies that mentioned the rate of at-will discharge or transfer to other hospital units, we considered these variables important in cases of patients with AAP because we observed meaningful differences between the groups. We believe that at-will discharging may negatively influence outcomes but that it is possible to lower the rate by well-tailored psychological interventions.

As in previous studies [17,18], we found that men were more prone to AAP. In contrast, research [19] on an Asian population found that, at the same drinking levels, women are more at risk of developing AAP than men. However, several general studies [20,21,22] on gender differences regarding binge drinking alcohol revealed that men are likelier to have higher alcoholism rates than women. Therefore, it is possible that our findings are likelier to stem from a higher prevalence of male alcoholism and a dose-dependent exposure. Indeed, in 2015, a meta-analysis [23] indicated a beneficial effect of moderate alcohol consumption (<40 g/day) in women that was not observed in male populations.

Our results in terms of age are congruent with the literature [24,25,26,27,28]; the median age of patients with AP ranges from 50 years [27] to 58 years [25] depending on the studied population and/or methodology. Notably, all relevant research comparing AAP with AP of other etiological causes (mainly biliary) has found that patients with AAP are either significantly younger [26,27,29] or that alcoholism is the main cause of AP in younger populations [24,28].

A prevalence of ICU admittance was found in our population of patients, similar to the findings of a large retrospective cohort study [27]. We discovered that patients with AAP are less likely to be admitted to the ICU, a finding similar to the results of Pu et al. [30], who compared AAP with biliary and hypertriglyceridemic AP, and Goyal et al. [15], who compared AAP with hypertriglyceridemic AP alone. Other researchers, such as Garcia et al. [31] and Balint et al. [32], found no differences pertinent to ICU admittance and AP etiology. However, a study from 2016 [33] revealed a higher rate of ICU admission in patients with AAP than in patients with OAP. All the divergent results might provide a glimpse into the heterogeneity of ICU rules of admission in terms of patients with AP; thus, we encourage their standardization. We did not see any statistically meaningful differences regarding LoS in the ICU.

We did not see any important differences concerning severity, as defined by the Revised Atlanta Classification [7]. Other investigators [2] found cases of AAP with milder severities. In a large retrospective cohort investigation conducted on an Asian population in 2017 [18], there were no differences among patients with AAP in terms of severity in the all-age population, although they found a higher incidence of SAP in the young and middle-aged groups.

We found a greater rate of pseudocysts in our AP population, similar to [33], and we found that AAP could be a risk factor for pseudocyst development. In a prospective study, Cui et al. [34] observed that AAP may be a risk factor for pancreatic fluid collection, and some of those collections will transform into pseudocysts. Szako et al. [35] showed that, compared with biliary etiology, alcoholic etiology is linked only with old pseudocysts.

Despite the small number of cases with WON in our cohort, a majority of them were associated with AAP. This matter necessitates prospective studies or at least other retrospective studies on a larger population, as other research teams have not found any links between the two [36].

Although we documented a lower cost of hospitalization in our population than other studies have reported [37,38,39], all other studies have been conducted in countries with far greater GDP per capita. However, we obtained a similar relation between AAP and costs as two studies [37,38], indicating that AAP is cheaper to treat than biliary AP. Although we did not directly compare AAP with biliary AP, most likely the majority of cases from OAP group were biliary.

We were unable to find comparisons regarding the type of ward preferred for patients during hospitalization, as there are no studies in the literature that compare this variable in terms of treating AAP. We are working on expanding our registry into a multicentric one to determine if this association is maintained in other centers as well.

Considering that our country has a nearly 1:1 ratio between urban and rural populations and that access to healthcare systems is poor in rural areas, we sought to identify an association between patients’ area of residence and AAP. However, no association was found. Some studies on this topic, mainly originating in China, found contradictory results: Pang et al. [40] noted a higher diagnosis rate of all-cause AP in suburban areas, but Fan et al. [41] observed a lower-than-expected prevalence of AAP in suburban areas. A study from New Zealand [42] found no differences between urban and rural origins regarding all-cause AP. Some research [43] has investigated socioeconomic deprivation in patients with AP, but a correlation between this issue and living in a rural area may be difficult to make and should be made with caution.

Our association between tobacco use and AAP is somewhat contradictory to some of the available literature. Multiple studies refer to smoking tobacco products as a risk factor for developing AP [19,44,45,46,47], and some of them refer specifically to AAP, but they could not find a correlation between smoking tobacco and AAP [19,45,47], instead regarding smoking mainly as an independent risk factor for all-cause AP. Still, one interesting study [46] associated smoking tobacco, heavy drinking and alterations to the PRSS1 and PRSS2 genes, which have been linked to developing AAP. Further investigation into the association between smoking and drinking in developing AP is needed. A limitation of our analysis on this subject comes from the low available data on this topic.

No correlation between etiology and LoS was observed in our cohort. However, we obtained a lower median LoS in patients with AAP (7 days) than Easler et al. [48] did (20 days), a similar one with O’Farrell et al. [49] and a longer one than Nesvaderani et al. [27] (4 days). Although Balint et al. [32] also they found no association, another study [48] concluded that AAP cases have longer LoS than all other etiologies.

Our study has several limitations, most of them residing in its retrospective design. Namely, we had missing data in regard to tobacco usage, low count of WON observed, there were some missing data on behalf of their medical history, and it was impossible to estimate the cumulated dose of exposure to ethanol in order to assess if there are any dose-dependent effects. Another limitation of our study resides in the fact that all the patients studied are originating from a single center, we are now working to expand our registry to other centers. The lack of CAGE questionnaire in assessing alcoholism might also hinder the results of our study.

## 5. Conclusions

We found that AAP is closely linked to lower ICU admittance and lower costs of hospitalization. The typical sketch of a patient suffering from AAP is a middle-aged male patient less likely to suffer from in-hospital mortality, has RAP, including complications like pseudocysts, has been treated in a gastroenterological ward and is discharged against medical advice. We stress that large prospective studies are required in this particular field in order to validate our results.

## Figures and Tables

**Figure 1 biomedicines-12-01299-f001:**
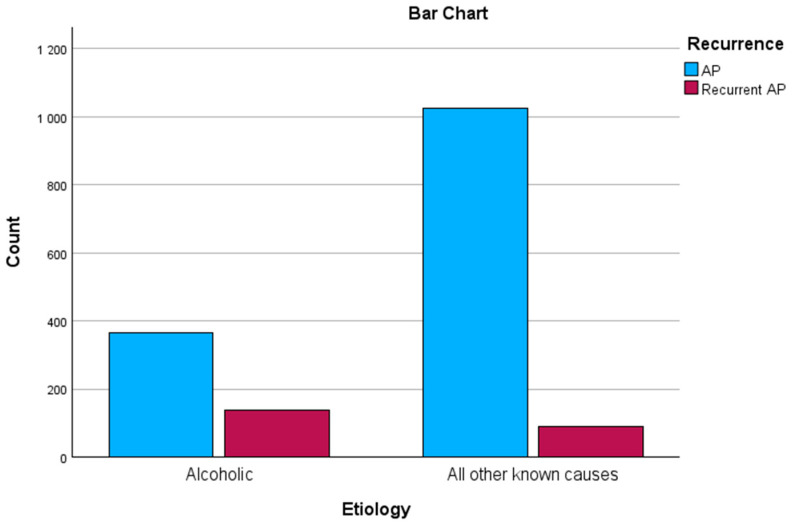
Recurrence related to etiology.

**Figure 2 biomedicines-12-01299-f002:**
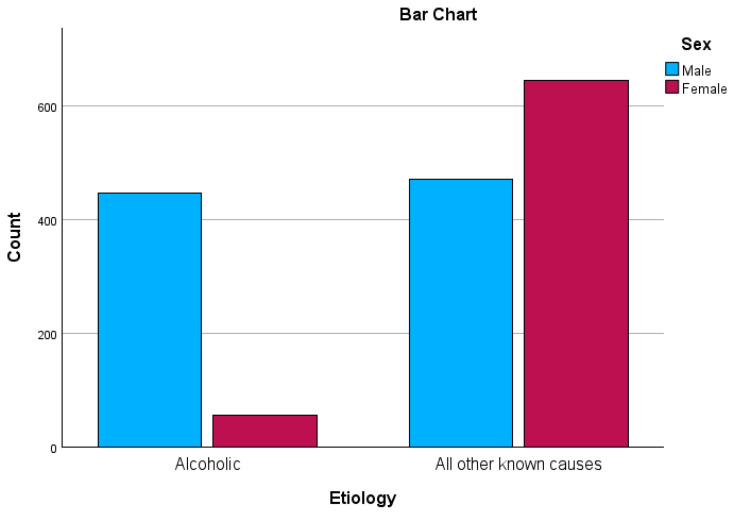
Gender distribution across groups.

**Figure 3 biomedicines-12-01299-f003:**
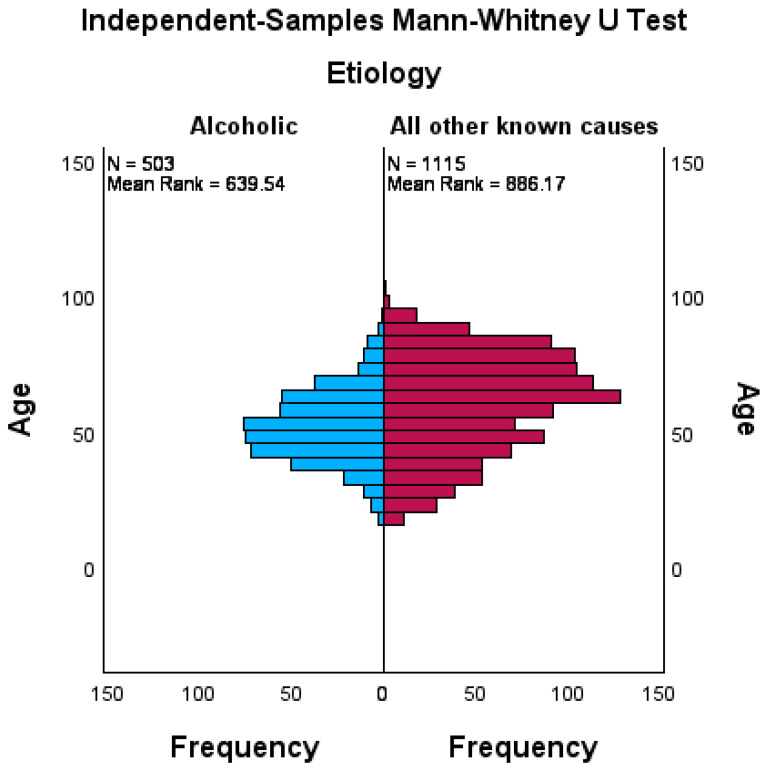
Age distribution in the two patient groups.

**Figure 4 biomedicines-12-01299-f004:**
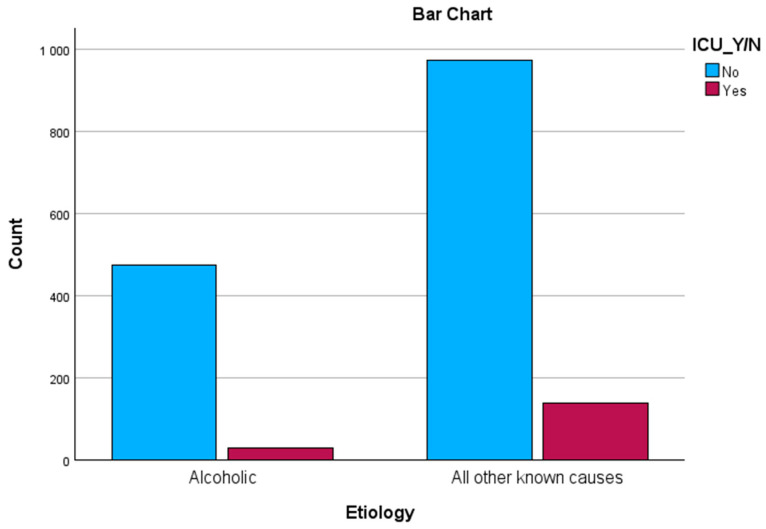
ICU admittance across groups.

**Figure 5 biomedicines-12-01299-f005:**
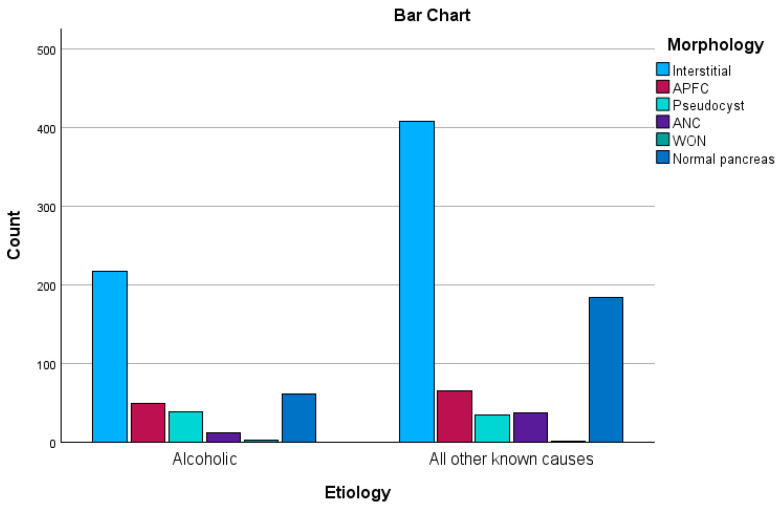
Distribution of morphology across groups.

**Table 1 biomedicines-12-01299-t001:** Registry population characteristics.

AP Patients (n = 1618)	
Recurrence	
Acute pancreatitis (first known attack)	1388 (85.8%)
Recurrent AP	230 (14.2%)
Age (years)	
Median	59.0 (IQR = 26.0)
Mean	57.7 (SD = 17.1)
Length of stay (LoS)	
Median	7.0 (IQR = 6.0)
Mean	8.8 (SD = 7.8)
Daily cost of hospitalization (EUR)	
Median	177.2 (IQR = 93.2)
Mean	419.4 (SD = 3475.8)
Etiology	
Alcoholic	503 (31.1%)
Idiopathic	253 (15.6%)
All other known causes	862 (53.3%)
Sex	
Male	918 (56.7%)
Female	700 (43.3%)
Severity	
Mild	821 (50.7%)
Moderate to severe	590 (36.5%)
Severe	207 (12.8%)
Morphology	
Interstitial	626 (38.7%)
Normal pancreas	245 (15.1%)
APFC	115 (7.1%)
ANC	50 (3.1%)
Pseudocyst	74 (4.6%)
WON	4 (0.2%)
No data	504 (31.1%)
Outcome	
Healed/ameliorated	1329 (82.1%)
Discharged at-will	96 (5.9%)
Deceased	108 (6.7%)
Transferred	74 (4.6%)
Stationary	11 (0.7%)
ICU admittance	
No	1449 (89.6%)
Yes	169 (10.4%)
Tobacco use	
Active	250 (15.5%)
Former (>4 weeks)	73 (4.5%)
Never	32 (2.0%)
No data	1263 (78.1%)
Ward of origin	
Gastroenterology	788 (48.7%)
Surgery	830 (51.3%)
Patient area of origin	
Urban	1159 (71.6%)
Rural	445 (27.5%)
Foreign	6 (0.4%)
No data	8 (0.5%)

**Table 2 biomedicines-12-01299-t002:** Clinical and demographic characteristics of patients with AAP and other known etiologies of AP.

	Alcoholic Etiology (n = 503)	Other Etiologies (n = 1115)	*p*-Value
Recurrence			
AP (first known attack)	364 (72.4%)	1024 (91.8%)	*p* < 0.01
Recurrent AP	139 (27.6%)	91 (8.2%)	
Sex			
Male	447 (88.9%)	471 (42.2%)	*p* < 0.01
Female	56 (11.2%)	644 (57.8%)	
Outcome at discharge			
Healed/Ameliorated	421 (83.7%)	908 (81.4%)	*p* < 0.01
Stationary	3 (0.5%)	8 (0.8%)	
Transferred	9 (1.8%)	65 (5.8%)	
Discharge at will	48 (9.6%)	48 (4.3%)	
Deceased	22 (4.4%)	86 (7.7%)	
Type of ward			
Gastroenterological	415 (82.5%)	373 (33.5%)	*p* < 0.01
Surgical	88 (17.5%)	742 (66.5%)	
ICU admittance			
Yes	29 (5.8%)	140 (12.6%)	*p* < 0.01
No	474 (94.2%)	975 (87.4%)	
Severity			
Mild	265 (52.7%)	556 (49.9%)	*p* = 0.28
Moderate to severe	183 (36.4%)	407 (36.5%)	
Severe	55 (10.9%)	152 (13.6%)	
Morphology			
Interstitial	218 (43.3%)	408 (36.6%)	
APFC	49 (9.7%)	66 (5.9%)	
Pseudocyst	39 (7.8%)	35 (3.2%)	*p* < 0.01
ANC	12 (2.4%)	38 (3.4%)	
WON	3 (0.6%)	1 (0.1%)	
Normal pancreas	61 (12.1%)	184 (16.5%)	
No data	121 (24.1%)	383 (34.3%)	
Urban or rural residency			
Urban	347 (69.0%)	812 (72.8%)	*p* = 0.23
Rural	147 (29.2%)	298 (26.7%)	
No data	9 (1.8%)	5 (0.5%)	
Tobacco usage			
Active	182 (36.2%)	68 (6.1%)	*p* < 0.01
Former	43 (8.5%)	30 (2.7%)	
Never	10 (2.0%)	22 (2.0%)	
No data	268 (53.3%)	995 (89.2%)	
Patient age (years)			
Median	51.0 (IQR = 18.0)	63.0 (IQR = 27.0)	*p* < 0.01
Mean	52.0 (SD = 13.0)	60.3 (SD = 18.1)	
Length of stay (days)			
Median	7.0 (IQR = 5.0)	7.0 (IQR = 6.0)	*p* = 0.99
Mean	8.7 (SD = 8.6)	8.8 (SD = 7.4)	
Daily cost of hospitalization (EUR)			
Median	154.7 (IQR = 62.0)	204.4 (IQR = 83.3)	*p* < 0.01
Mean	311.1 (SD = 2878.9)	552.4 (SD = 4029.1)	

## Data Availability

Data available upon reasonable request from the corresponding author. According to Romanian and EU laws publishing them can be considered a privacy data breach and as such we can not make them publicly available.

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
