# Peer review of "Alcoholic Acute Pancreatitis, a Retrospective Study about Clinical Risk Factors and Outcomes—A Seven-Year Experience of a Large Tertiary Center"

_biomedicines, 2024, doi:10.3390/biomedicines12061299_

Round 1

Reviewer 1 Report

Comments and Suggestions for Authors

The authors analyze epidemiological data of acute pancreatitis (AP) patients in Bucharest. They compare clinical characteristics, outcome, length of hospital stays and even costs of treatment of acute alcoholic pancreatitis (AAP) to the other etiologies. The subject of paper is of interest for the clinicians and the very high proportion of AAP in Romania permits an adequate analysis. 

Criticism and questions:

 ·       The authors write in the introduction:   “Worldwide, of all cases of AP that end in mortality, estimated mortality rates of40.2% and 12.3% among men and women with AAP, respectively, have been reported. The highest mortality rate related to AAP has been reported in Russia (2.7/100,000 people) [4]” (lines 38-40): These two numbers are completely different: the first numbers are related to the lethality of AAP and they are extremely high, the authors should precise the source of these numbers, which are far from their own experiences. The second number is an epidemiological aspect concerning the population-based mortality published from Russia.

 ·       As detailed in “Methods”, the authors – correctly – excluded the cases with pancreatic cancer. However, there is a confusion with the diagnosis acute-on-chronic pancreatitis. ((see lines 75-77).

 ·       Alcoholic etiology was established with interviews with the patients. Who conducted these interviews? Somebody of the authors or other medical doctors? It could be difficult to establish clearly and objectively the alcoholic etiology. From one part, in Central Europe the alcohol consumption is very frequent and can result in overestimation of its role, but from the other side, patients - mainly women - frequently deny the existent regular alcohol consumption

 ·       The authors consider moderately decompensated diabetes mellitus as etiology of AP. It is not usual, they should precise literature which support this consideration. In a relatively recent review on the relation of diabetes and AP (Richardson A, Park WG: Acute pancreatitis and diabetes mellitus: a review. Korean J Intern Med 2021; 36:15-24 https://doi.org/10.3904/kjim.2020.505), the authors do not mention any role of DM in the etiology of AP. This level of decompensation is certainly insufficient to explain AP. There are several published cases of severe acute necrotizing pancreatitis with diabetic ketoacidosis, but in these cases diabetes is considered as a consequence and not as etiology. However, 55 patients in this paper were considered to have AP as a consequence of decompensated DM, very similar to hypertriglyceridemia. It is a considerable number of cases. Anyway, these criteria require citation of literature and solid arguments which support this unhabitual affirmation. 

 ·   “Trauma" as etiology, includes ERCP?

 ·       The numbers are not clear enough: e.g. the authors probably consider as 100% the 2042 patients remaining after the exclusion of miscoded and duplicate cases, but this number does not appear in the paper. 

 ·       The proportion of AAP is very high, but it is usual in this geographic region. 

·   The authors obtained information about tobacco smoking in a minority of the cases, but 73% of these patients were active smokers, not 18.6%.  In addition, they found significantly more active smokers in alcoholic group.

 ·   AAP was more frequently recurrentIt is an expected finding, but the objective demonstration is of interest. However, we can interpretate this finding, that the lack of resolution of known etiological factors favors the recurrence. E.g. : how much patients of biliary AP continued to have unresolved biliary pathology at the moment of recurrence? Or on the contrary: if an alcoholic patients stopped drinking completely, did he have a recurrent acute attack?

 ·   108 patients died, 179 were admitted to UCI. How much of these 179 died?  Could the authors analyze the mortality in the three groups of severity?

 ·   “…we observed a lower in-hospital mortality rate among patients with AAP (see Table 2), although no statistically significant difference was found.” This difference was not statistically significant. Thus, it is more correct: there was no significant difference in mortality. 

 ·   The authors observed only 4 patients with WON, all of them in the AAP group. It is probably the consequence of the small number:  necrotic pancreatitis of any origin can produce WON

 ·   It would be easier to understand the costs in euro or in USD. Calculating with 1 Euro = 5 RON, the costs were really considerably lower than the majority of published data. 

Reviewer 2 Report

Comments and Suggestions for Authors

Thanks for submitting this manuscript. The greatest strength of this manuscript lies in the authors' mention that it includes the largest number of pancreatitis patients in the country to date. However, there are several shortcomings:

1. The authors did not use standard three-line tables, and it is recommended that the tables in the current manuscript be revised accordingly.

2. The authors listed multiple hospitalizations for the same patient as separate cases, possibly introducing a potential statistical bias into the study results. It is suggested that this be corrected and that the statistical results be provided based on different patients as separate cases.

3. The authors' analysis of the limitations of the retrospective study was overly concise, for example, omitting that the retrospective study was a single-center study and that the current research content is relatively simple.

Comments on the Quality of English Language

Minor editing of English language required

Round 2

Reviewer 1 Report

Comments and Suggestions for Authors

The authors answered the majority of my questions and doubts, and considerable modifications were made in the manuscript, first of all, the presentation of data was changed. The problem of diabetes as etiology of AP does not exist anymore, being absent from the new version. Equally, the absence of post ERCP from the list of etiologies remained without importance. However, some questions remained unresolved. 

“Compared with OAP the in-hospital mortality rate was statistically significant lower, so AAP might be considered a protective factor related to this particular outcome” (lines 234-35). While one can understand that the authors refer to their own observations (minor mortality in alcoholic pancreatitis than in the rest of etiologies), one cannot speak about the “protective role” of alcohol, being the most important etiology of the disease in the authors’ country. 

The authors discuss their findings on tobacco use in a whole paragraph. However, they obtained information only in the minority of their patients, this proportion does not permit a valuable discussion. I suggest to reduce this paragraph: The lack of information in the majority of our patients does not permit any conclusion. 

The same is true about the cases complicated with WON, while in this case the problem is not the lack of information.

Finally, the "conclusions" could be improved

Reviewer 2 Report

Comments and Suggestions for Authors

Thanks for submitting the revised manuscript, which has shown a noticeable improvement compared to the initial draft.

Comments on the Quality of English Language

Minor editing of English language required
